# Evolving Mechanisms in the Pathophysiology of Pemphigus Vulgaris: A Review Emphasizing the Role of Desmoglein 3 in Regulating p53 and the Yes-Associated Protein

**DOI:** 10.3390/life11070621

**Published:** 2021-06-26

**Authors:** Ambreen Rehman, Yunying Huang, Hong Wan

**Affiliations:** 1Centre for Oral Immunobiology and Regenerative Medicine, Institute of Dentistry, Barts and The London, School of Medicine and Dentistry, Queen Mary University of London, London E1 2AT, UK; ambreen.rehman@duhs.edu.pk (A.R.); yunying.huang@qmul.ac.uk (Y.H.); 2Department of Oral Diagnosis and Medicine, Dr Ishrat Ul Ebad Khan Institute of Oral Health Sciences, Dow University of Health Sciences, Karachi 74200, Pakistan

**Keywords:** Pemphigus Vulgaris, desmoglein-3, cell signaling, skin blistering disease, disease pathogenesis

## Abstract

The immunobullous condition Pemphigus Vulgaris (PV) is caused by autoantibodies targeting the adhesion proteins of desmosomes, leading to blistering in the skin and mucosal membrane. There is still no cure to the disease apart from the use of corticosteroids and immunosuppressive agents. Despite numerous investigations, the pathological mechanisms of PV are still incompletely understood, though the etiology is thought to be multifactorial. Thus, further understanding of the molecular basis underlying this disease process is vital to develop targeted therapies. Ample studies have highlighted the role of Desmoglein-3 (DSG3) in the initiation of disease as DSG3 serves as a primary target of PV autoantibodies. DSG3 is a pivotal player in mediating outside-in signaling involved in cell junction remodeling, cell proliferation, differentiation, migration or apoptosis, thus validating its biological function in tissue integrity and homeostasis beyond desmosome adhesion. Recent studies have uncovered new activities of DSG3 in regulating p53 and the yes-associated protein (YAP), with the evidence of dysregulation of these pathways demonstrated in PV. The purpose of this review is to summarize the earlier and recent advances highlighting our recent findings related to PV pathogenesis that may pave the way for future research to develop novel specific therapies in curing this disease.

## 1. Introduction

The autoimmune disorder, Pemphigus is a blistering disease that results in intraepithelial blisters affecting predominantly the mucosa and skin. It is mediated by circulating pathogenic autoantibodies that attack the keratinocyte cell surface proteins leading to acantholysis [1]. Acantholysis indicates the loss of intercellular connection between keratinocytes in the epidermal stratum spinosum that renders the intraepidermal clefts. Two common subtypes of pemphigus with distinct clinical and immunopathological features are characterized, i.e., Pemphigus Vulgaris (PV) and Pemphigus Foliaceus (PF) with PV comprises of approximately 70% of the pemphigus cases. Furthermore, there are two subsets of PV; the mucosal dominant form is caused by autoantibodies targeting desmoglein-3 (DSG3) only and the mucocutaneous form has the autoantibodies targeting both desmoglein-1 (DSG1) and DSG3.

Clinically, PV presents as multiple flaccid blisters limited to the suprabasal layer in the epidermis or mucosal membrane, whilst the keratinocytes in the top layers maintain their cell cohesion. Interestingly, the basal cells also maintain their adhesion to the basement membrane, thereby histologically appearing like a ’row of tombstone’ [2]. On the other hand, in PF, autoantibodies target DSG1 that leads to acantholysis in the granular layer of the epidermis sparing the oral mucosa and clinically presents as superficial frail blisters [3,4]. If left untreated, PV is fatal due to the failure of the skin barrier that culminates in fluid loss, electrolyte disturbances and contracting the opportunistic infection. The most common treatment modality is the administration of oral steroids and immunosuppressive agents and so far, there is still a lack of specific treatment for these diseases.

The desmosomal cadherin DSG3 was first identified as an isoform of the desmoglein subfamily and the auto-antigen in PV [5]. Following this pioneer finding, the pathogenicity of anti-DSG3 autoantibodies in PV has been confirmed by ample in vivo and in vitro studies. Anhalt et al. injected purified IgGs from PV patients into the skin of neonatal mice and showed induction of cutaneous blisters in more than seventy per cent of mice, in a dose-dependent manner, compared to normal control IgGs which showed no signs of such blistering [6]. Histological examination revealed the blister location between the basal and immediate suprabasal layers in the epidermis where DSG3 resides, resembling the acantholysis in patients with PV. This finding was confirmed by other independent studies that demonstrate that the PV IgG induced blisters are DSG3 specific since DSG3 deficient cells fail to show the loss of intercellular connection when treated with PV IgG [7].

DSG3 have been indicated as a key mediator involved in desmosome remodeling, epidermal proliferation and differentiation, cell migration and apoptosis, and thus validating that DSG3 acts as a signaling molecule that has a major impact on tissue integrity and homeostasis. The binding of PV IgG to DSG3 has been demonstrated to stimulate the phospholipase C signaling pathway which causes calcium release and PKC activation leading eventually to the desmosome and cell surface DSG3 depletion and the loss of intercellular cohesion and formation of blisters [8]. Later studies have identified that PV IgG targeting DSG3 activates p38 MAPK, Src and Rho A [9,10,11].

An in vivo transgenic mouse study has shown increased keratinocyte proliferation and altered terminal differentiation associated with misexpression of Dsg3 in the suprabasal layers of the skin suggesting this basal cell isoform of DSG is involved in regulating cell proliferation [12]. Findings from our group support this notion by showing that DSG3 depletion in HaCaT keratinocytes caused suppression of cell proliferation and colony growth [13]. Such an effect of the altered DSG3 expression on cell proliferation indicated a link between DSG3 loss and cell cycle control [14] but the molecular mechanism was largely unknown till recently [15].

Owing to the widespread expression of DSG3 at the plasma membrane beyond the desmosomes it is thought that the non-junctional pool of DSG3 may be responsible for cell signaling and is the primary target of PV IgG [16]. A previous study by our group has demonstrated that non-junctional DSG3 regulates Src via interaction with classical E-cadherin and helps regulate the adherens junction formation [17,18], suggesting the DSG3′s broader role in cell biology. In accordance with this notion, other studies have independently implicated that DSG3 plays a role in cancer as it is found to be upregulated in squamous cell carcinomas (SCCs) [19] although the function of DSG3 in tumor cell biology remains an area poorly defined. This present review will briefly summarize the current understanding of pemphigus pathology with a focus on our recent findings of DSG3′s novel signaling roles in the regulation of p53 and YAP that have important implication in pemphigus acantholysis.

## 2. Pathogenesis of PV

The pathological mechanisms of PV are controversial and remain a hot topic under debate in the field. Ample studies support the concept that PV is caused by the pathogenic IgG autoantibodies directed against DSG3 as well as DSG1. However, there is evidence also suggesting pemphigus blistering is not solely caused by autoantibody targeting to DSG3 but rather results from intracellular signaling mediated by non-DSGs targeting autoantibodies that leads to shrinkage of cell and apoptosis [20,21,22]. Nevertheless, the pathological mechanisms of blister formation in PV are still not clearly understood [20,23]. Therefore, various theories have been proposed to explain the pathogenesis of PV, including the following.

### 2.1. Steric Hindrance of Cell Adhesion Caused by Autoantibodies

This long-standing theory states that the binding of autoantibodies to the extracellular domain of DSGs causes loss of intercellular adhesion and pemphigus acantholysis [5,24,25]. However, other reports have identified that this is insufficient to produce intercellular adhesion loss, and hence signaling (inside-out) is considered to be important to trigger keratinocyte loss of contact in PV [9,26].

### 2.2. Desmoglein Compensation

The different distribution patterns of DSG1 and DSG3 in the skin and oral mucosa support this theory. It describes that if both DSG1 and DSG3 are present and only one is inactivated by autoantibodies then the other will compensate and provide adhesion, but if only one DSG is present and is attacked by the autoantibodies, then the acantholysis will occur [3]. However, accumulating data suggest that this theory is unable to justify the disease presentation of PV. Blistering in patients with autoantibodies against both DSG1 and DSG3 should not only be restricted to suprabasal level but rather should appear across the entire epidermis. Nonetheless, this does not seem to happen. Moreover, some cutaneous PV patients show no DSG1 titer, while other mucosal PV patients are found lacking the anti-DSG3 antibodies. Furthermore, there is concern about several other cadherins that also are expressed in keratinocytes. Why can’t they compensate for the loss of DSG3 and DSG1? Therefore, these findings and concerns cannot solely be explained by the desmoglein compensation theory [27,28].

### 2.3. Outside-In Signaling

Various signaling pathways are at play in the development of pemphigus acantholysis. The phosphorylation of DSG3 is caused by the binding of pathogenic PV IgGs to DSG3 followed by its dissociation from plakoglobin and then endocytosis-mediated degradation [29]. PKCα shift from the cytoplasmic fraction to the membrane in cultured human keratinocytes after exposure to PV IgG was first demonstrated by Osada et al., suggesting that PKC isomers with different activation profiles may be important in modulating intracellular signaling events triggered by PV IgG binding to DSG3 [30]. Subsequently, PV IgG treatment of primary human keratinocytes activates epidermal growth factor receptor (EGFR), a mechanism that is downstream of p38 MAPK [31]. Thus, this finding demonstrates p38 MAPK guided interaction between DSG3 and EGFR. Deregulation of other signaling pathways such as Src, Rho GTPases as well as actin disorganization have also been implicated in PV [19]. Desmosomes remodel regularly to allow epidermal cell migration during keratinocyte differentiation and wound healing by switching between two adhesive modes i.e., “calcium-independent hyper-adhesion” and “calcium-dependent weak-adhesion”. PKC, Src, and EGFR signaling facilitate this process [32,33,34,35]. Disruption of this remodeling process including the activation of the above-mentioned signaling pathways has been reported in PV [24,31]. This is followed by DSG3 internalization that is facilitated by p38 MAPK activation, resulting in the DSG3 exhaustion from the desmosomes, a specific event in PV, leading to pemphigus acantholysis [24].

### 2.4. Antibody-Induced Cell Shrinkage and Apoptosis Involved in the Activation of the p53 Pathway

This theory explores new mechanisms with the findings obtained by independent groups. Early studies have proposed that apoptosis may be responsible for pemphigus acantholysis [36,37]. The apoptosis pathway in PV is proved by experimental studies that demonstrate that apoptotic signaling is activated by PV autoantibody and anti-Fas receptor (FasR) antibody. According to the studies, soluble Fas ligand (FasL) is secreted first, and then levels of FasR, FasL, Bax, and p53 are raised within the cell. As a result, Bcl-2 levels are depleted, whereas caspase-8, caspases 1/3, and death-inducing signaling complex are activated [38,39,40,41,42]. Autoantibody mediated apoptosis was inhibited as a result of depletion of caspases 1 or 3, effectively preventing the process of acanthosis, supporting the idea that apoptosis contributes to cell dissociation [36,42,43].

In 2009, Grando et al. proposed a new terminology ‘apoptolysis’- *basal cell shrinkage hypothesis*, to explain the pathogenesis of acantholysis. After the pathogenic PV autoantibodies bind to the keratinocyte receptor(s), a series of signal-transduction pathways trigger the rupture of the cytoskeleton, resulting in the collapse and shrinkage of the keratinocytes. This theory also vividly links to cell apoptosis and basic pathological features, emphasizing that apoptotic enzymes contribute to acantholysis development in terms of both molecular events and chronologic sequence [21]. Although the existence of apoptosis in pemphigus has been reported with a view that this is associated with the surface receptors other than DSGs [21,44], there was still a lack of study with a focus on the p53 pathway in PV pathogenesis and thus, the link between DSG3 and p53 was not established till our recent study.

Rehman et al. have identified DSG3 acting as an anti-stress protein by counterbalancing p53 in the maintenance of normal epithelial homeostasis and demonstrated disruption of this regulation in PV [15] (Figure 1). The study addresses the long-term controversy in the field of pemphigus research and fills the gap for the link between DSG3 and p53. It indicates that the loss/disruption of DSG3 alone is sufficient to induce aberrant p53 response and thus the disruption of DSG3 mediated intercellular junctions does not result solely from steric hindrance but rather causes dysfunction of cellular anti-stress response. This finding uncovers a previously unidentified biological function of this desmosomal cadherin in keratinocytes and suggests that blisters are partly due to the altered stress response that acts in the orchestra with p53 resulting in the damage of normal tissue homeostasis. Hence, disruption of this pathway may play an important part in PV pathology.

Some signaling pathways that are activated in PV [9,45,46] are the known elements in the p53 signaling network such as p38 MAPK and c-Myc [47,48]. Hence, the activation of p53 in PV could be via some of these pathways downstream of antibody targeting DSG3, e.g., p38 MAPK. Studies have shown that PV IgG enhances the p38 MAPK mediated apoptosis in keratinocytes [49] and the inhibitors of p38 MAPK can suppress the stimulation of the proapoptotic proteinase caspase-3 [50]. Hence, these results suggest that the p38 MAPK activation may be involved in the p53 mediated apoptosis in PV leading to blistering. Studies also show that acantholysis occurs prior to DSG3 depletion [51] and in addition, EGFR seems to be another factor in p38 MAPK signaling and apoptosis triggered by PV IgG. Grando et al. showed the activation of EGFR as a critical step in the initiation of PV IgG signaling [21]. Passive transfer of pathogenic AK23 in an adult mouse model demonstrated the activation of EGFR before any signs of DSG3 depletion and blister formation [52]. In line with this finding, Bektas et al. demonstrated that suppression of EGFR limited DSG3 internalization and keratin retraction triggered by PV IgG [31]. Increased expression of c-Myc downstream of EGFR activation was also demonstrated [52] as well as in another independent study by Williamson et al. [46]. Thus, both p38 MAPK and c-Myc activation could participate in the DSG3/p53 pathway leading to apoptosis and blistering in PV. c-Myc is also known to regulate apoptosis via post-translational regulation of p53 stability [53]. Interestingly, Pusapati et al. demonstrated that the p53 induction in response to c-Myc overexpression requires ATM kinase, a major regulator of the cellular response to DNA double-strand breaks (DSBs) [54]. The study by Rehman et al. showed the evidence of DSBs and upregulation of ATM kinase in DSG3 depleted cells, along with upregulation of the p53 pathway [15]. Collectively, it is likely that PV autoantibodies targeting DSG3 affect its function in the anti-stress network that leads to oxidative/antioxidative imbalance in cells (discussed below). As a consequence, oxidative stress can occur in PV cells resulting in DSBs and hence the activation of a chain of reactions including ATM, c-Myc and p38 MAPK, and subsequently evokes the p53 pathway leading to deregulation of the apoptotic machinery and PV blistering (Figure 1).

It is worth noting that the Dsg3/p53 pathway in keratinocytes established by Rehman et al. seems to be unique. To address the specificity of this pathway, Rehman et al. have performed the RNAi knockdown experiments on other junctional proteins, such as desmoplakin (a marker of desmosomes) and E-cadherin (the classical cadherin in adherens junctions). They found that neither desmoplakin knockdown nor E-cadherin knockdown gave rise to p53 induction (even when exposed these siRNA transfected cells to UVB) similar to what had been observed in the DSG3 knockdown cells [15]. Furthermore, the authors also performed a double knockdown for DSG3/p53 and confirmed the attenuation of the p53 target p21^WAF1/CIP1^ in these double knockdown cells compared to the respective controls [15]. These findings suggest that the DSG3/p53 pathway is indeed DSG3 specific and imply an exclusive role for Dsg3 acting as a stress sensor and responder in keratinocytes beyond its adhesive function in desmosomes. Thus, deregulation of this pathway in PV contributes to pemphigus acantholysis (Figure 1). However, this finding does not rule out the activation of the p53 pathway in other diseases since p53 is a central player in cellular stress response. Different signaling inputs or molecular cascades can consequently trigger the p53 activation. For instance, bullous pemphigoid (BP) is a closely related autoimmune blistering disease where basal keratinocytes lose the attachment to the basement membrane, leading to sub-epithelial bullae [55]. This disease is caused by autoantibodies against two hemidesmosome antigens BP180 and BP 230 [55,56]. Indeed, enhanced p53 was detected in BP as well as in cultured keratinocytes treated with BP sera (our unpublished data). Although both PV and BP are caused by autoantibodies targeting the cell junctional proteins (desmosomal and hemidesmosomal, respectively), we believe distinct pathophysiologies occurred in these two conditions though some overlapping features may exist in p53 activation [57]. In accordance with this notion, abnormal mitochondrial function and ROS overproduction are reported in both PV and BP [58].

Reports about the clinical association of PV with cancer are few and vary. Several earlier studies indicated a favorable effect of DSG3 reduction in carcinogenesis with its decrease correlated with a more aggressive cancer phenotype [59,60,61]. In addition, aberrant expression of Dsg3 is also reported in basal cell carcinoma (BCC) where its expression correlates with Dsg2 at the mRNA level but differ greatly at the protein level, with the loss of coordination of DSG2 and DSG3 expression and a reduction of DSG3 in BCC [62,63]. Nevertheless, paradoxically, a potential role of DSG3 as an oncogene in cancer development has been emerging. In this regard, several investigations have indicated the possible contribution of DSG3 in carcinogenesis [64,65] and have considered DSG3 as a positive biomarker in SCC [19]. In line with this notion, immunohistochemistry of DSG3 in cutaneous SCC showed enhanced positivity, with the signals positively correlated with poor differentiation status of cancer (our unpublished data). The seminal findings of Rehman et al study also suggest a pre-cancerous role for DSG3 by hindering the tumor suppressor p53 [15]. This notion is further supported by in vitro studies of the group that indicate that overexpression of DSG3 in A431 carcinoma cells promotes cell migration and invasion via the mechanisms of enhanced signaling of Src, Rac1/Cdc42, Ezrin and c-Jun-AP-1 transcription factor [17,18,64]. However, this area of study is still in its infancy. This is partly due to the lack of transgenic animal models with overexpression of DSG3 that will enable the study of its role in carcinogenesis and cancer spreading. Further work is required for characterization of this pathway in tumor cell biology, essential to advance our knowledge about DSG3 in cancer and the development of potential anti-cancer therapy by manipulating DSG3, such as the usage of specific anti-DSG3 antibodies that spares DSG3 function in cell adhesion.

Taken together, the above-discussed studies link together several unrelated pathways and open new avenues of research to investigate the pathogenesis and stress-induced tissue damage in PV. It also suggests a potential new direction of novel approaches in the diagnosis and treatment of not only PV but also other diseases including cancer.

### 2.5. Oxidative Stress-Mediated YAP Dysregulation

Increased accumulation of reactive oxygen species (ROS) or free radicals can lead to oxidative stress. Although ROS is indispensable at moderate concentrations to defend from microorganisms involved in the cellular signaling systems [66], excess ROS can lead to opposite detrimental effects such as DNA damage, lipid peroxidation, dysregulated signal transduction, altered cellular function and even cell death [67,68]. DNA damages such as DSBs are extremely toxic to cells and can cause diverse human diseases including cancer. Lipid peroxidation can adversely affect membrane structural integrity and function where various intercellular junctions including adherens junctions and desmosomes reside. Protein oxidation renders their carbonylation, an irreversible post-translation modification that inactivates proteins. Cells possess a self-defense mechanism to combat various reactive oxygen species. Antioxidants are the key components in this system and protect the cells against the toxic effects of excess free radicals via changing them to harmless forms (H_2_O, O_2_, etc.) or preventing their formation. The imbalance of oxidant/antioxidant is identified in various autoimmune diseases including rheumatoid arthritis and lupus erythematosus [69,70,71,72,73,74]. However, there remain limited studies regarding oxidative stress in PV and in keratinocytes treated with PV IgG though the evidence obtained suggests this is the case [75,76,77,78,79,80] (Table 1). Reduction of antioxidants, including their transcripts and antioxidant capacity, has been detected in PV patient blood samples and in addition, increased oxidative stress is found to be correlated to clinical disease activity [75,79]. However, it remains not completely clear whether the oxidant/antioxidant disequilibrium is the cause or effect of PV IgG-mediated disruption of cell-cell adhesion.

**Figure 1 life-11-00621-f001:**
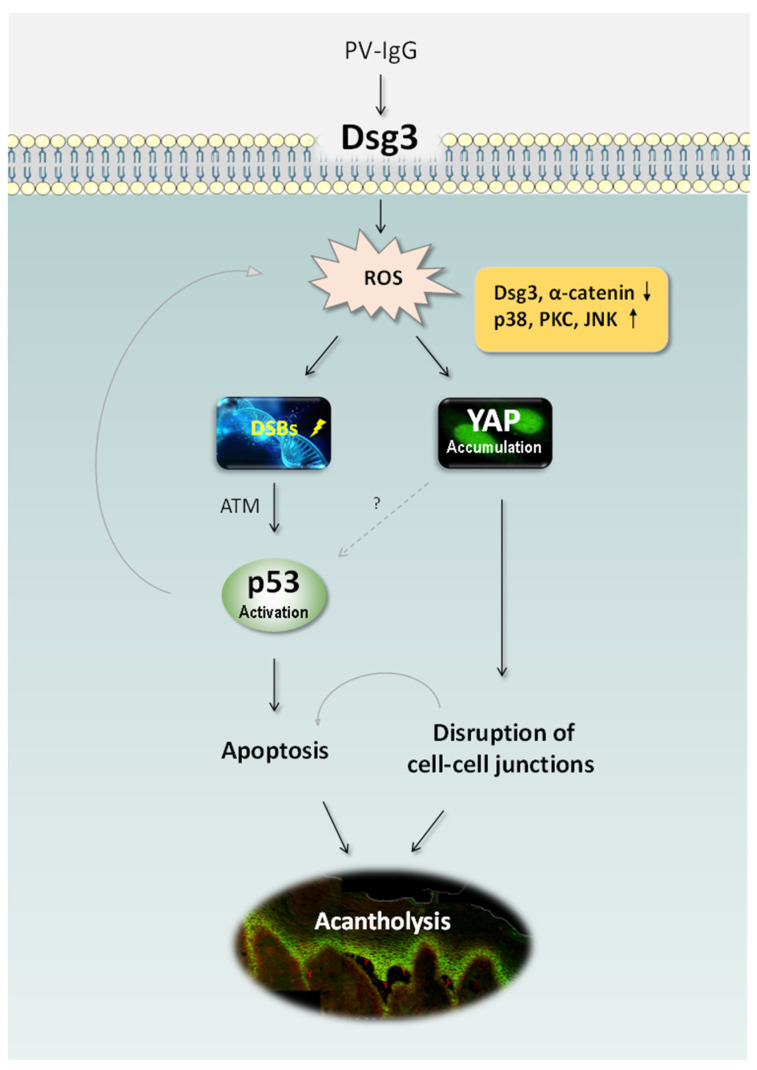
Schematic representation of the recently identified pathways in PV [15,81]. PV IgG targeting DSG3 perturbs its action in anti-stress response and causes elevated ROS generation, leading to DNA DSBs, p53 activation via ATM and consequently apoptosis. ROS overproduction also induces YAP dysregulation leading to the reduced expression of adhesion proteins including DSG3 and α-catenin and ultimately disruption of cell-cell adhesions. Oxidative stress can also trigger the activation of various MAPK pathways such as p38, PKC and JNK resulting in YAP accumulation. The synergistic effects of apoptosis and cell-cell dissociation can cause the clinical manifestation of blistering in PV. It remains unknown whether YAP dysregulation has any impact on p53 in PV. ATM: Ataxia-Telangiectasia Mutated, DSBs: double-strand breaks, PKC: protein kinase C, JNK: c-Jun N-terminal kinase, ROS: reactive oxygen species, YAP: Yes-associated protein.

A recent study by our group has demonstrated that elevated ROS production can be evoked by treating keratinocytes with PV patient’s sera as well as a pathogenic mouse monoclonal antibody AK23, with concomitant accumulation of Yes-associated protein (YAP) especially in the cytoplasm of antibody-treated cells [81]. Importantly, a marked increase of YAP was also detected in the oral mucous membrane of PV patients with elevated cytoplasmic and/or nuclear YAP found in cells not only surrounding blisters but also in the perilesional regions where the split of keratinocytes was not shown. YAP is a crucial downstream co-transcription factor in the Hippo pathway and also is involved in the cellular anti-stress response in Hippo-dependent and –independent manners. Additionally, elevated YAP was demonstrated in cells exposed to hydrogen peroxide that mimics cellular oxidative stress, with numerous protein aggregates located at the cell borders. Furthermore, the effect of YAP on cell junction formation and maintenance was investigated and showed that expression of exogenous YAP led to disruption of cell junctions, whereas an adverse effect was observed in cells with YAP silencing that enhanced DSG3 expression which was capable of abrogating AK23-mediated pathogenicity to DSG3 [81]. These findings collectively suggest that antibody-induced oxidative stress and cytoplasmic YAP accumulation may attribute to the disruption of cell junctions and pemphigus acantholysis (Figure 1). Many stress signals such as serum and glucose starvation, PKA activation, disruption of the actin cytoskeleton, and Src inhibition, etc. are capable of inducing YAP cytoplasmic localization and inhibition of its transcription activity that involve p38 MAPK activation [82]. In addition, YAP activation has also recently been demonstrated by heat shock in a Hippo kinase-dependent manner [83]. Significantly, our study showed that treating cells with antioxidants, such as glutathione (GSH) and N-acetyl cysteine (NAC), as well as the inhibitors for various MAPK pathways including p38 MAPK, was able to suppress hydrogen peroxide-induced YAP coupled with enhanced cell-cell adhesion. Taken together, these results suggest that oxidative stress-mediated YAP dysregulation likely plays a key role in pemphigus blistering, indicating the potential therapeutic role of antioxidants in the treatment of PV. p53 can act as a key coordinator of oxidative stress when its expression levels are high [84] and thus p53 stabilization and activation in PV (as described above) can accelerate oxidative stress and ROS overproduction leading to senescence and cell death.

Another recent study by our group has determined that DSG3 is capable of regulating YAP independent of antibody binding to DSG3 [85,86]. RNA interference-mediated DSG3 silencing in keratinocytes resulted in YAP reduction and downregulation of its target genes accompanied by repression of cell proliferation as indicated by Ki67 staining compared to control cells, the finding that is consistent with our previous report [13]. These results prompted us to hypothesize that DSG3 positively regulates YAP. Paradoxically, elevated YAP was shown in PV with variations of its cellular localization. Increased YAP nuclear expression was pronounced in PV perilesional regions [81] suggesting activation of YAP likely occurs at the early stages of the disease process when the autoantibodies target DSG3 and/or other surface proteins. However, the different effects of monoclonal versus polyclonal IgGs, as well as PV IgG targeting non-DSGs, on ROS and YAP expression remain largely unknown. It is possible that the alterations of YAP in PV are complicated and involved multiple mechanisms associated with oxidative stress which warrants further investigation.

### 2.6. Non-Desmoglein Antibodies

Although autoantibodies against DSGs are essentially the cause for the disease initiation, many non-DSG antibodies have also been implicated in blister formation in PV [87,88]. For instance, autoantibodies against muscarinic acetylcholine receptors (mAChRs) have been found in pemphigus patient’s sera [87]. These receptors regulate cell adhesion and motility and thus their inactivation by antibodies can lead to keratinocyte detachment and eventually acantholysis [87]. Anti-mitochondrial antibodies (AMA) are also detected in PV patients [88]. AMA can trigger an intrinsic apoptotic pathway by causing cytochrome C release that leads to the activation of caspase-9 aiding acantholysis [88]. Dysfunction of the mitochondrial pathway can also trigger ROS overproduction leading to cellular oxidative stress. Both plakophilin-3 (PKP3) and plakoglobin belong to the Armadillo-repeat family of adhesion proteins [89]. Autoantibodies against PKP3 is demonstrated in PV patients [90]. PKP3 is involved in desmosome assembly and stabilization and mice deficient in this gene showed altered desmosome and adherens junctions [91]. Plakoglobin reduction induces c-Myc and p38 MAPK accumulation resulting in keratinocyte disadhesion [46,92]. Plakoglobin is shown to be linked to the antigenic complex that is targeted by antibodies in PV [93]. Sera of PV patients, including those with acute PV having negative detection of anti-DSG1/3 autoantibodies, also contain antibodies against the classical E-cadherin [94] as well as several other adhesion molecules such as DSG4 [95,96], desmoplakin [97], desmocollins [90], PERP [90], and pemphaxin [98]. A recent review has described approximately fifty proteins expressed in keratinocytes, including adhesion molecules, receptors, enzymes and mitochondrial components can be the targets of PV autoantibodies [99]. Patients with sera lacking anti-DSG antibodies can still develop typical clinical and histological features of pemphigus. Furthermore, a study by Chernyavsky et al. has demonstrated that autoantibodies with preabsorption for anti-Dsc3, anti- M3AR (M3 muscarinic acetylcholine receptor) or anti- SPCA1 (secretory pathway Ca^2+^/Mn^2+^-ATPase isoform 1) can prevent blister formation in the neonatal mouse skin of both the passive transfer model and skin explant [100]. However, this study also showed that each of these immunoaffinity-purified autoantibodies alone failed to induce a PV-like skin lesion in neonatal mice unless the mixture of these purified autoantibodies were administered. Hence, the synergistic role of these antibodies in PV pathogenesis is highly advocated [90,100,101]. Although numerous structural and metabolic proteins that regulate keratinocyte adhesion and maintenance have been identified to contribute to the pathological process of PV, their distinct roles in the disease initiation have not been proved and significant work is needed to determine the pathogenicity of these autoantibodies.

### 2.7. T-Cell Dysregulation

The pathological mechanism in PV is triggered by autoantibodies and the immune system is involved in the disease’s onset and progression. Various studies have suggested an increased risk of pemphigus in those with a family history of other autoimmune disorders, as well as the presence of specific genotypes associated with HLA and non-HLA genes [102,103]. To date, PV and HLA class II genes association is advocated strongly and published widely [102,104,105]. T cells are critically involved in both the effector and regulatory immune responses and have been demonstrated to have a variety of defects that contribute to PV immunopathogenesis. Th1 and Th2 cytokine concentrations are altered in PV compared to healthy controls that lead to immune function impairment [106]. T helper 17 (Th17) levels in PV patients are elevated, compared to healthy individuals [107,108] but its role in PV pathogenesis is not established. A compensatory response in an attempt to maintain epithelial homeostasis was suggested. The dysfunction of regulatory T cells (Tregs) is reported in patients with PV [108,109]. Treg cells prevent autoimmune response by controlling antigen mediated inflammation and immune response. An in vivo study demonstrated anti-Dsg3 antibody production is controlled by Tregs [110]. Furthermore, induction of Tregs in the PV HLA-transgenic mouse model reduces anti-Dsg3 autoantibodies [111]. The above-discussed studies highlight the pivotal role of T cells in PV immunopathogenesis. Decreased peripheral tolerance and increased autoantibody production by B cells are elicited by altered function and morphology of T cells that subsequently results in inflammation and immune cell infiltration, ultimately leading to the disease manifestations.

## 3. Conclusions

The pathological mechanisms underlying the pemphigus acantholysis are complex and remain not completely clear. Over the past three decades, significant advances have been made in our understanding of PV pathophysiology. Autoantibodies targeting the N-terminus of DSG3 are likely the initial and critical step that has an impact on its action in anti-stress response and hence triggers various intracellular events leading to blistering in tissues expressing DSG3 that include skin and oral mucosa. The direct inhibition of DSG3′s trans-interactions is not sufficient to induce the complete loss of cell cohesion and the synergistic action of orchestra signaling networks such as p38 MAPK, EGFR, Src, c-Myc, and Rho GTPases are considered to be vital in the pathophysiology of PV. Likely, the non-DSG antibodies also play a role, however, it remains unclear if they are responsible for the PV clinical phenotype [87,90]. Additionally, genetic predisposition to PV is well-reported and T cells are demonstrated as a critical player in autoimmunity. Moreover, an increased incidence of apoptotic occurrence has been implicated in PV acantholysis lesions. Elevated apoptotic factors such as Bax, FasL, and caspases as well as downregulation of the apoptotic inhibitory protein Bcl-2 in PV have been reported. Our recent studies shed new light on the unprecedented roles of DSG3 and suggest its mediated signaling towards p53, YAP and ROS in cellular stress response with the evidence of dysregulation of these axes being demonstrated in PV [15,81,85,86,112]. Elucidation of the underlying mechanisms could be a fertile area for future research [113]. Moreover, the finding of oxidative stress-mediated YAP dysregulation in PV [81] opens a new avenue to explore a novel therapeutic approach with antioxidants. Thus, this review lays the basis for a better understanding of the PV pathogenesis that may help in the development of novel diagnostic tools and treatment of this life-threatening autoimmune disease in the future.

## Figures and Tables

**Table 1 life-11-00621-t001:** Summary of oxidative stress research in PV.

Origins	Sample Size	Components Detection	Oxidant/Antioxidant Imbalance	Conclusions	References
Plasma & RBC	18	Levels of lipid peroxidation and antioxidants	**↑** Oxidant/**↓** Antioxidant (Vitamin A & E, β-carotene, CAT, GSH-Px, GSH)	Oxidant/Antioxidant imbalance predominantly in favor of pro-oxidant side, and/or diminished TAC or levels of antioxidants	[75]
Plasma	30	Levels of antioxidants	Oxidant/**↓** Antioxidant (Uric acid)		[77]
Serum	27	Levels of TOC, LOOH, TAC OSI = TOC/TAC	**↑** Oxidant (TOC, LOOH, OSI)/Antioxidant		[76]
Serum	47	Levels of TAC	Oxidant/**↓** Antioxidant (TAC)		[79]
Plasma & Serum	43	Levels of CAT, GSH-Px, TAC	**↑** Oxidant (CAT, GSH-Px)/**↓** Antioxidant (TAC)		[80]
Serum	9	Levels of PC and TAC	**↑** Oxidant (PC)/**↑** Antioxidant (TAC)	Only study shown oxidant/antioxidant imbalance with enhanced TAC	[78]
Keratinocytes	8	ROS levels in cells treated by PV sera or AK23	**↑** Oxidant (ROS)	Oxidant/Antioxidant imbalance with ROS overproduction evoked by PV sera or AK23	[81]

Abbreviation: CAT, catalase; GSH-Px, Glutathione Peroxidase; GSH, Glutathione; TOC, Total oxidant capacity; LOOH, Lipid hydroperoxide; TAC, Total antioxidant capacity; OSI, Oxidative stress index; PC, Protein carbonyl; ROS, Reactive oxidative species.

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
