# Peer review of "Evolving Mechanisms in the Pathophysiology of Pemphigus Vulgaris: A Review Emphasizing the Role of Desmoglein 3 in Regulating p53 and the Yes-Associated Protein"

_life, 2021, doi:10.3390/life11070621_

Round 1
Reviewer 1 Report
- I suggest changing the title into: Evolving mechanisms in the pathophysiology of pemphigus vulgaris: A review emphasizing the role of desmoglein 3 in regulating p53 and the yes-associated protein. It better reflects the content of the manuscript.
- The is no need to use capitals in full names of diseases.
- I suggest replacing the word primitive with seminal in the sentence: The primitive findings of Rehman et al study suggest a pre-cancerous role for Dsg3 by hindering the tumour suppressor p53. This sentence lacks ref. number.
- I suggest adding a reference suggesting a role for DSG3 in BCC that found discordant expression of DSG2 and DSG3 at the mRNA and protein levels in nodular and superficial basal cell carcinoma revealed by immunohistochemistry and fluorescent in situ hybridization (https://doi.org/10.1111/ced.12355) to the last section of the paragraph 2.4.
- I am just curious whether the events meticulously described in this comprehensive review are pemphigus vulgaris-specific or not. To cast light on this important issue, I would suggest adding the description of what is going on within the keratinocyte downstream after, let’s say, antibody-BP180 target epitope and antibody- BP230 target epitope bindings in bullous pemphigoid (BP). This would serve as an apt comparison as BP shares with pemphigus vulgaris autoimmune pathology. If this is impossible to do, I suggest that authors should explicitly state and document their opinion on whether the described intrakeratinocyte events in pemphigus vulgaris are indeed pemphigus vulgaris-specific or not. My gut feeling is that there are not, as numerous diverse stimuli trigger the same intracellular pathways.
- It would be better to use the following abbreviations throughout the manuscript: DSG (human protein), Dsg (animal protein), DSG (in italics) (human gene), Dsg (in italics) (animal gene). Such style is recommended by the biological community.
Author Response
We thank the reviewer for the expert comment. Now we have made the changes throughout the document as suggested by the reviewer. Please see our point-by-point response to the reviewer in the attached file.

Reviewer 2 Report
This is an excellent review of desmoglein signaling in pemphigus. This is an important and under-discussed concept. The review is thorough and well organized.
I only have two minor comments below.
I think the following sentences can be removed. The discussion of differential Dsg expression in cancer is important, but the introduction regarding relationship between cancer in PV is somewhat confusing. The presence of PV with underlying cancer (quite rare and inconsistently found) doesn't really tie in to direct expression of Dsg in SCC (which is more pertinent to the article)
"Pemphigus blistering also occurs in association with neoplasms in a
condition known as paraneoplastic pemphigus (56). An increased incidence of internal malignancy in PV patients has been reported in the literature (57, 58). "
I would also recommend reference to the following primary study which highlights incidence of non-desmoglein antibodies PLoS One. 2013;8(3):e57587
Author Response
We thank the expert reviewer's comment and have made changes accordingly. Please also see our point-by-point response in the attached file.
